

# 1 Effective hydraulic properties of 3D virtual stony soils identified
# 2 by inverse modeling

Mahyar Naseri[*], Sascha C. Iden, and Wolfgang Durner
Division of Soil Science and Soil Physics, Institute of Geoecology, Technische Universität Braunschweig, Germany
*Corresponding author:* m.naseri@tu-bs.de, Langer Kamp 19c, 38106 Braunschweig, Germany
s.iden@tu-bs.de, Langer Kamp 19c, 38106 Braunschweig, Germany
w.durner@tu-bs.de, Langer Kamp 19c, 38106 Braunschweig, Germany
**Core Ideas**
▪ Virtual stony soils with different rock fragment contents were generated in 3D using the Hydrus 2D/3D
software.
▪ Evaporation experiments and unit-gradient experiments were numerically simulated.
▪ We used inverse modelling with the Richards equation to identify effective hydraulic properties of virtual
stony soils.
▪ The identified hydraulic properties were used to evaluate the scaling models of calculating hydraulic
properties of stony soils.
**Keywords**
Soil hydraulic properties, water retention curve, hydraulic conductivity, stony soil, inverse modeling, Hydrus 2D/3D
**Abstract**
Stony soils that have a considerable amount of rock fragments are widespread around the world. However,
experiments to determine effective hydraulic properties of stony soils (SHP), i.e. the water retention curve (WRC) and
hydraulic conductivity curve (HCC), are challenging. Installation of measurement devices and sensors in these soils
is difficult and the data are less reliable because of high local heterogeneity. Therefore, effective properties of stony
soils especially in unsaturated hydraulic conditions are still not well understood. An alternative approach to evaluate



the SHP of these systems with internal structural heterogeneity is numerical simulation. We used the Hydrus 2D/3D
software to create virtual stony soils in 3D and simulate water flow for different volumetric rock fragment contents,
f. Soils with volumetric stone contents from 11 to 37 % were created by placing impermeable spheres in the form of
rock fragments in a sandy loam soil. Time series of local pressure heads in various depths, mean water contents and
fluxes across the upper boundary were generated in a virtual evaporation experiment. Additionally, a multi-step unit
gradient simulation was applied to determine effective values of hydraulic conductivity near saturation up to pF=2.
The generated data were evaluated by inverse modeling, assuming a homogeneous system, and the effective hydraulic
properties were identified. The effective properties were compared with predictions from available scaling models of
SHP for different volumes of rock fragments. Our results showed that scaling the WRC of the background soil based
on only the value of f gives acceptable results in the case of impermeable rock fragments. However, the reduction of
conductivity could not be simply scaled by the value of f. Predictions were highly improved by applying the Novák,
Maxwell, and GEM models to scale the HCC. The Maxwell model matched the numerically identified HCC best.

## 1. Introduction

Stony soils are soils with a considerable amount of rock fragments and are widespread in mountainous and forested
watersheds around the world (Ballabio et al., 2016; Novák and Hlaváčiková, 2019). Rock fragments in soil are
particles with an effective diameter of larger than 2 mm (Tetegan et al., 2015; Zhang et. al., 2016). Their existence in
soil influences the two constitutive soil water relationships known as soil hydraulic properties (SHP) i.e. water
retention curve (WRC), and hydraulic conductivity curve (HCC) (Russo, 1988; Durner and Flühler, 2006). The
accurate identification of SHP is a prerequisite for adequate prediction of water flow in soil with the Richards equation
(Farthing and Ogden, 2017; Haghverdi et al., 2018). The SHP depend on soil texture and structure (Kutilek, 2004;
Lehmann et al., 2020), and are influenced by the presence of rock fragments in soil. It is generally accepted that rock
fragments decrease the water storage capacity of soils and its effective unsaturated hydraulic conductivity. In contrast,
the formation of macropores in the vicinity of embedded rock fragments may lead to an increase in saturated hydraulic
conductivity. While experimental evidence and theoretical analyses show that the volumetric content of rock
fragments, $f$ ($m^3$ $m^{-3}$), has the highest influence on effective SHP of stony soil, the effect of other characteristics of
rock fragments such as their porosity, shape, size, arrangement, and orientation towards flow is less clear (Hlaváčiková
and Novák, 2014; Hlaváčiková et al., 2016; Naseri et al., 2020). Up to the present, two approaches have been dominant
in identifying the hydraulic behavior of stony soils: I) Experimental setups with the aim of measuring SHP of stony



soils in the field or in controlled systems in the laboratory (Cousin et al., 2003; Dann et al., 2009; Grath et al., 2015;
Beckers et al., 2016, Naseri et al., 2019), and II) Development of empirical, physical or physico-empirical approaches
to scale hydraulic properties of background soil based on the volumetric content of rock fragments ($f$) and their
characteristics (Novák et al., 2011; Naseri et al., 2020). These two approaches have some systematic limitations that
restrict their applications in investigating the hydraulic behavior of stony soils. Installation of sensors and
measurement instruments in the stony soils are technically demanding (Cousin et al., 2003; Verbist et al., 2013,
Coppola et al., 2013; Stevenson et al., 2021), undisturbed sampling is laborious (Ponder and Alley, 1997), relatively
larger samples are required (Germer and Braun, 2015), and the measured data might be more inconsistent due to the
higher local heterogeneity of such soils (Baetens et al., 2009; Corwin and Lesch, 2005). Furthermore, some of the
available scaling models to obtain effective SHP are conceptually oversimplified and they exclusively consider the
volume of rock fragments as the only input parameter (Bouwer and Rice, 1984; Ravina and Magier, 1984).
Additionally, they assume impermeable rock fragments and are proposed mainly for saturated flow conditions. These
scaling models need a systematic verification under variably-saturated conditions using experimental data or 3D
simulations. Some reviews of these models and their evaluation are available in the literature (Brakensiek et al., 1986;
Novák et al., 2011; Beckers et al., 2016; Naseri et al., 2019).
Hlaváčiková and Novák (2014) proposed a model to scale the HCC of the background soil, parametrized with the van
Genuchten–Mualem (van Genuchten, 1980) model, using the model of Bower and Rice (1984). Hlaváčiková et al.,
(2018) used the water content of rock fragments as input parameter to scale the WRC of the background soil. Naseri
et al. (2019) used the simplified evaporation method (Peters et al., 2015) to experimentally determine the effective
SHP of small soil samples containing various amounts of rock fragments. Their study criticizes the application of the
scaling models developed for saturated stony soils to unsaturated conditions and emphasizes the need to develop
approaches that consider more characteristics of the rock fragments to calculate SHP of the stony soils.
Recent advancements in computational hydrology and computing power suggest the numerical simulation of soil
water dynamics as a promising alternative to the measurement of effective SHP of heterogeneous soils (Durner et al.,
2008; Lai and Ren, 2016; Radcliffe and Šimůnek, 2018). Numerical simulations have several advantages. They do not
demand strict experimental setups, are repeatable under a variety of initial and boundary conditions, and in contrast
to the laboratory experiments, space and time scales are not restrictive factors in the simulations. These assets have



made them a favorable tool in water and solute transport modeling in heterogeneous soils (Abbasi et al., 2003;
Šimůnek et al., 2016). However, with few exceptions, heterogeneous soils like stony soils have been simulated only
for simplified cases, i.e., either under fully saturated conditions or with reduced dimensionality, i.e., simulations of
stony soils in two spatial dimensions (2D). Novák et al. (2011) calculated effective saturated hydraulic conductivity
($K_s$) of soils containing impermeable rock fragments using steady-state simulations with the software Hydrus 2D
which solves the Richards equation in two spatial dimensions. They derived a linear relationship between the $K_s$ of
stony soil and $f$. Hlaváčiková et al. (2016) simulated different shapes and orientations of rock fragments in Hydrus
2D to obtain the effective $K_s$ of the virtual stony soils. Beckers et al. (2016) used Hydrus 2D simulations to extend the
investigations towards the impact of different volumetric rock fragment contents, shape, and size on the HCC. They
also identified effective SHP of a silt loam soil containing rock fragments using laboratory evaporation experiments
for rock fragment contents up to 20 % (v/v).
The inverse modeling approach has been applied to identify effective hydraulic properties of soils in laboratory
experiments (Ciollaro and Romano, 1995; Hopmans et al., 2002; Nasta et al., 2011), in lysimeters and field (Abbaspour
et al., 1999; Abbaspour et al., 2000), virtual lysimeters with internal textural heterogeneity (Durner et al., 2008; Schelle
et al., 2013), and WRC of stony soils through field infiltration experiments (Baetens et al., 2009). Although theoretical
studies and laboratory investigations on packed samples are insufficient to understand fully the hydraulic processes in
stony soils, they do lead the way to the improvement and validation of effective models and their application at the
field and even larger scales. Inverse modeling is arguably the best approach to achieve these aims because it allows to
validate effective models using process modeling. Our aim in this study was to investigate the application of inverse
modeling to identify the effective SHP of 3D virtual stony soils and to explore its applicability to these soil systems
as an example of internal structural heterogeneity. We were interested in answering the following questions:
i) Is it possible to describe the dynamics in the heterogeneous 3D system with the 1D Richards equation assuming a
homogeneous soil?
ii) If so, what are the effective SHP of stony soils and how are they related to the SHP of the background soil?
To answer these questions we conducted forward simulations of water movement in 3D using the Richards equation
as variably-saturated flow model. We created stony soils by embedding voids representing impermeable spherical
rock fragments as inclusions into a homogeneous background soil. Then we simulated transient evaporation



experiments and stepwise steady-state, unit-gradient infiltration experiments in 3D. The generated data were used as
an input to a 1D inverse model to obtain the effective SHP of stony soils, and these properties were used to evaluate
and compare the available scaling models of SHP for stony soils.

## 2. Materials and methods

### 2.1. Simulation model

The Hydrus 2D/3D software was used to generate virtual stony soils and simulate the water flow in the created three-
dimensional geometries. Water flow in Hydrus 2D/3D is modelled by the Richards equation (Šimůnek et al., 2006;
2008), which is the standard model for variably-saturated water flow in porous media. The Hydrus 2D/3D software
solves the mixed form of the Richards equation numerically using the finite-element method and an implicit scheme
in time (Celia et al., 1990; Šimůnek et al., 2008; 2016; Radcliffe and Šimůnek, 2018). The three-dimensional form of
the Richards equation under isothermal conditions, without sinks/sources, and assuming an isotropic hydraulic
conductivity is:

$$\frac{\partial \theta}{\partial t} = \frac{\partial}{\partial x}\left[K(h)\left(\frac{\partial h}{\partial x}\right)\right] + \frac{\partial}{\partial y}\left[K(h)\left(\frac{\partial h}{\partial y}\right)\right] + \frac{\partial}{\partial z}\left[K(h)\left(\frac{\partial h}{\partial z} + 1\right)\right] \tag{1}$$

where $\theta$ is the volumetric water content (cm$^3$ cm$^{-3}$), $t$ is time (s), $h$ is the pressure head (cm), and $K(h)$ is the hydraulic
conductivity function (cm d$^{-1}$). $x$, and $y$ (cm) are the horizontal Cartesian coordinates, and $z$ (cm) is the vertical
coordinate, positive upwards. We used the van Genuchten-Mualem model to parametrize the WRC and HCC (van
Genuchten, 1980):

$$S_e(h) = \frac{\theta(h) - \theta_r}{\theta_s - \theta_r} = [1 + (\alpha h)^n]^{-m} \tag{2}$$

and

$$K(h) = K_s S_e^\tau \left[1 - \left(1 - S_e^{\frac{1}{m}}\right)^m\right]^2 \tag{3}$$



where $\theta_s$ and $\theta_r$ are the saturated and residual water contents (cm$^3$ cm$^{-3}$), respectively, $S_e(h)$ is the effective saturation
(-), $\alpha$ (cm$^{-1}$) is a shape parameter, $n$ is an empirical parameter related to the pore size distribution (-) and $m = 1 -$
$1/n$, $K_s$ is the saturated hydraulic conductivity and $\tau$ is a tortuosity/connectivity parameter (-).

### 2.2. 3D geometries representing stony soils

The virtual stony soils in 3D were created by placing spherical inclusions in a background soil. In accordance with
real laboratory experiments (not reported here), we generated virtual soil columns as cylinders with a diameter of 16
cm and a height of 10 cm and an total volume of $\approx$2011 cm$^3$. The inclusions were considered as voids representing
impermeable rock fragments embedded in the background soil. Configurations and characteristics of the created 3D
geometries of stony soils are illustrated in Fig. 1. Each spherical inclusion had a diameter of 3.04 cm and a volume of
$\approx$14.7 cm$^3$. Stony soils with different volumetric rock fragment contents were created by including different numbers
of spherical inclusions in the soil column. A total number of 15, 27, 39, and 51 spherical inclusions in each column
led to four volumetric rock fragment contents of 11.0, 19.8, 28.5 and 37.3 % (v/v). Spheres were arranged in the
column in three layers. The spheres' centers were in depths of 2.5, 5.0 and 7.5 in the column and each layer was
packed with one-third of the total number of intended spheres. Furthermore, observation points at selected nodes of
the numerical grid were inserted in each of the three depths of the column (i.e. 2.5, 5.0 and 7.5 cm) in the background
soil and not in close vicinity of the inclusions to provide time series of soil water pressure head for the inverse
simulations. For the background soil, a homogenous sandy loam soil was considered with the van Genuchten-Mualem
model parameters $\theta_s = 0.410$ (cm$^3$ cm$^{-3}$), $\theta_r = 0.065$ (cm$^3$ cm$^{-3}$), $\alpha = 0.01$ (cm$^{-1}$), $n = 2.0$ (-), $\tau = 0.5$ (-), and $K_s =$
100 (cm d$^{-1}$). The targeted mesh size for the different simulations was set to 0.25 cm. The dependency of the numerical
solution on the mesh size was tested with some refined meshes and negligible differences in the results were obtained
for different mesh sizes.




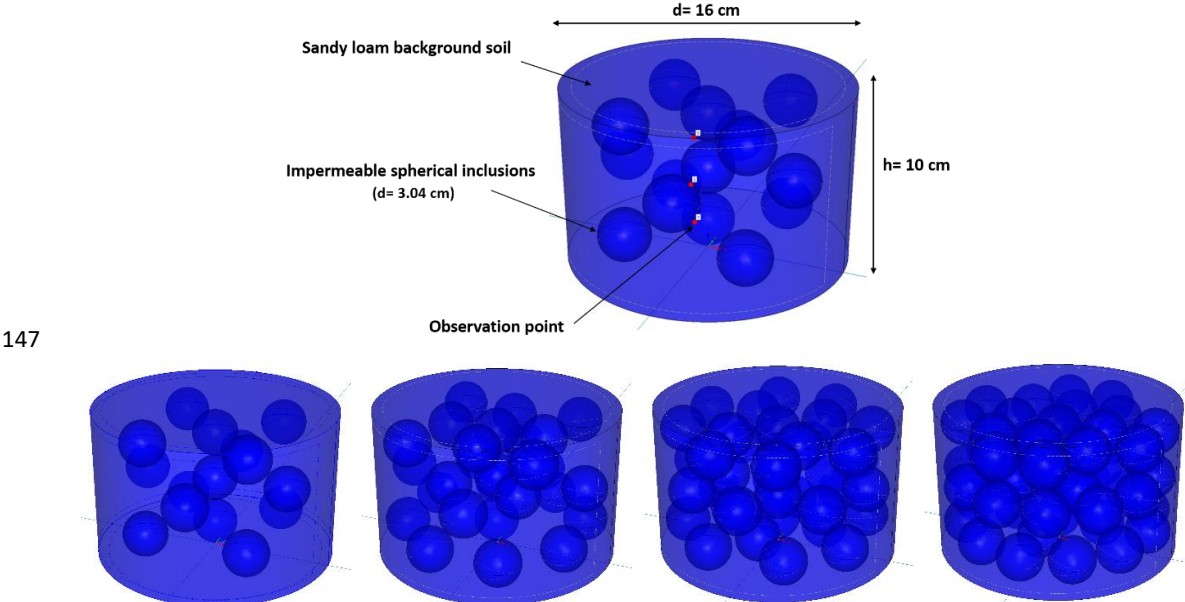


*Figure 1: Visualization of 3D configurations of the generated stony soils including the dimension of rock fragments and soil cylinder and location of the observation points (top). Bottom row shows rock fragment contents of 11.0, 19.8, 28.5 and 37.3 %,*

*from left to right.*

### 2.3. Forward simulations

We simulated evaporation (EVA) (Peters and Durner, 2008) and multistep unit gradient (MSUG) experiments (Sarkar
et al., 2019). For EVA, a linear distribution of pressure head (-2.5 cm top, +7.5 cm bottom) was used as the initial
condition. The boundary conditions were no-flux at the bottom and atmospheric with a constant potential evaporation
rate of 0.6 (cm d$^{-1}$) and zero precipitation at the top. The EVA experiments were simulated for 10 days and the time
series of pressure heads at each observation point, the initial volumetric water content and the cumulative evaporation
and evaporation rate were collected for later use in the inverse simulations.
In the MSUG experiment, the soil column was initially fully saturated with a constant pressure head of 0 cm. A
sequence of step-wise decreasing constant pressure heads was assigned to the upper and lower boundaries of the
column. The duration of the MSUG was 100 days and the pressure head in the upper and lower boundaries was
simultaneously decreased stepwise to a pressure head of -100 cm. The applied pressure heads $h_i$ were 0, -1, -3, -10, -
20, -30, -60, and -100 cm, respectively. Time steps were chosen such that a steady-state flow condition was reached



for each pressure step, indicated by identical water fluxes at the top (inflow) and bottom (outflow) boundaries and
constant pressure heads at the observation points. The hydraulic conductivities at the respective pressure heads $h_i$ data
were calculated by dividing the steady-state water flux rates (cm$^3$ d$^{-1}$) by the total surficial area of the soil column ($\approx$
202 cm$^2$).
The converging and diverging flow field around obstacles produces even under unit gradient conditions spatially
different pressure heads, and as opposed to saturated conditions, these different pressure heads are under unsaturated
conditions associated with different water saturations and different local hydraulic conductivities. We were interested
in whether and to what extent this could lead to nonlinear effects in the derivation of the effective hydraulic properties,
in particular the effective HCC. Furthermore, since the flow field for a given volume fraction of obstacles depends on
dimensionality, i.e., is different in a 2D simulation than in a 3D simulation, studying the effects in the unsaturated
region was one of the main motivations for performing this numerical analysis in 3D.

### 2.4. Inverse modeling of evaporation in 1D

A 10-day evaporation experiment in 1D was simulated with the software package HYDRUS-1D (Šimůnek et al., 2006;
2008) to obtain the SHP parameters using inverse modeling. The generated data from the EVA and MSUG forward
simulations in 3D were used as an input to the 1D inverse simulations. Time series of the pressure heads at three
observation depths, mean volumetric water contents in the column during the EVA experiment, and the data points of
the effective HCC from the MSUG experiment were used as data in the objective function. The time series of the
mean volumetric water content was calculated from the initial water content, cumulative evaporation and soil volume.
The measurement range for pressure heads used in the objective function was from saturation down to -2000 cm. This
reflects a setup with laboratory tensiometers with boiling delay (Schindler et al., 2010). The time series of the
simulated evaporation rates from the 3D simulations were used as the time variable atmospheric boundary condition
for the 1D inverse simulations. The 1D soil profile was 10 cm long and was discretized into 100 equally sized finite
elements. Similar to the 3D simulations, three observation points were defined in the depths of 2.5, 5.0 and 7.5 cm. A
no-flux boundary condition was used at the bottom. The six parameters of the van Genuchten model occurring in Eq.
(2) and (3) were all simultaneously estimated by inverse modeling. The weighted-least-squares objective function was
minimized by the SCE-UA algorithm (Duan et al., 1992). The data obtained from the EVA experiment allows to
identify the WRC from saturation to the pressure where the tensiometers fail, and the HCC in the mid to dry range of





the SHP (roughly -100 to -2000 cm pressure head), while the MSUG provides a precise determination of the HCC in
the wet range (Sarkar et al., 2019; Durner and Iden, 2011). As evaporation experiments do not provide information on
hydraulic conductivity near water-saturation (Peters et al., 2015), we included the MSUG data in the object function
for the inverse simulation of the EVA experiments to improve the uniqueness of the inverse solution and the precision
of the identified HCC near saturation (see Schelle et al., 2010, for another example).

## 2.5. Predicting SHP of stony soils by scaling models

The SHP of stony soils obtained by inverse modeling were compared to SHP that are predicted by available scaling
models and used for their evaluation. Considering that the volumetric content of rock fragments has the dominant
influence on the WRC of a stony soil, a common approach is partitioning the WRC and HCC of stony soil based on
the volume of each component in the soil-rock mixture and calculating the effective SHP of stony soil using the
volume averaging or the composite-porosity model. The general form of the WRC model considers the moisture
contents of the background soil $\theta_{\text{soil}}$ (h) (cm$^3$ cm$^{-3}$) and embedded rock fragments $\theta_{\text{rock}}$ (h) (cm$^3$ cm$^3$) to calculate the
effective WRC of stony soils $\theta_{\text{m}}(h)$ (cm$^3$ cm$^{-3}$) (Flint and Childs, 1984; Peters and Klavetter, 1988) with the following
form in the full moisture range (Naseri et al., 2019):

$$\theta_{\text{m}}(h) = f\theta_{\text{rock}} + (1 - f)\theta_{\text{soil}} \qquad (4)$$

A typical assumption in stony soils hydrology is that the porosity of rock fragments is negligible. In this case, Eq. (4)
reduces to (Bouwer and Rice, 1984):

$$\theta_{\text{m}}(h) = (1 - f)\theta_{\text{soil}} \qquad (5)$$

For the effective hydraulic conductivity of stony soils, some scaling models are developed for saturated conditions
that might apply to the hydraulic conductivity at any pressure heads. The simplest scaling model accounts only for the
reduction in the cross-sectional area available for flow of water. This leads to the equation (Ravina and Magier, 1984):

$$K_{\text{r}} = 1 - f \qquad (6)$$

where $K_{\text{r}}$ (-) is the relative hydraulic conductivity of stony soil, i.e., $K_{\text{r}} = K_{\text{m}}/K_{\text{soil}}$, where $K_{\text{m}}$ is the effective hydraulic
conductivity of the stony soil (cm d$^{-1}$), and $K_{\text{soil}}$ is the conductivity of the background soil (cm d$^{-1}$).



Ina more recent approach, Novák et al. (2011) developed a linear relationship based on the 2D numerical simulation
results as a first approximation to scale the saturated hydraulic conductivity of stony soils:

$$K_\mathrm{r} = 1 - \alpha f \qquad (7)$$

The parameter $\alpha$ was reported to depend on the texture of the background soil, with a range between 1.1 for sandy
clay to 1.32 for clay. This model is easy to apply, but it requires the estimation of the parameter $\alpha$ to calculate $K_\mathrm{r}$. For
our calculations, we assumed $\alpha = 1.2$ for the sandy loam background soil used in our study.
Another model that has been developed for mixtures with spherical inclusions is the Maxwell model (Maxwell, 1873;
Corring and Churchill, 1961; Peck and Watson, 1979; Zimmermann and Bodvarsson, 1995). It takes the volumetric
rock fragment content, hydraulic conductivity of the background soil and hydraulic conductivity of inclusions into
account to calculate the hydraulic conductivity of the stony soil. In the special case of impermeable inclusions,
Maxwell model reduces to:

$$K_\mathrm{r} = \frac{2(1 - f)}{2 + f} \qquad (8)$$

A recently developed model by Naseri et al. (2020), which is based on the general effective medium theory (GEM),
allows considering effects of permeability, shape, and orientation of rock fragments on the effective HCC. For
impermeable rock fragments, the GEM model reduces to the following form:

$$K_\mathrm{r} = \left(1 - \frac{f}{f_\mathrm{c}}\right)^t \qquad (9)$$

where $f_\mathrm{c}$ is the critical rock fragment content with values between 0.84 and nearly 1 and $t$ is a shape parameter with
values between 1.26 and nearly 1.5 for spherical rock fragments. In this study, we set the critical rock fragment to
$f_c = 0.982$ (v/v) according to the size ratio of the rock fragments to the background soil, and $t = 1.473$ for spherical
rock fragments (for details, see appendix in Naseri et al., 2020).
It should be noted that all approaches apply at any pressure head $h_i$, i.e., the scaling that is originally developed for
saturated conditions with locally constant hydraulic conductivity in the background soil is equally applied to
unsaturated conditions.



### 3. Results and discussion

#### 3.1. Flow field and variability of state variables in the MSUG experiment

Figure 2 visualizes the pressure head (cm), water content ($cm^3 \; cm^{-3}$), and velocity ($cm \; d^{-1}$) in a 2D cross section in the center of the soil column through the forward simulation of the MSUG experiment. The profile is shown for the steady state flux situation with a pressure head of -100 cm and the stony soil with 28.5 % rock fragment content. The Figure shows a considerable change in the flow velocities, even at the upper boundary. Also, as Fig. 2 illustrates, the conditions above an obstacle might be slightly wetter than below an obstacle, but the variations in the pressure head and the water content fields is very small. We note that this general finding was equally applicable for all other pressure heads steps in the MSUG experiment.





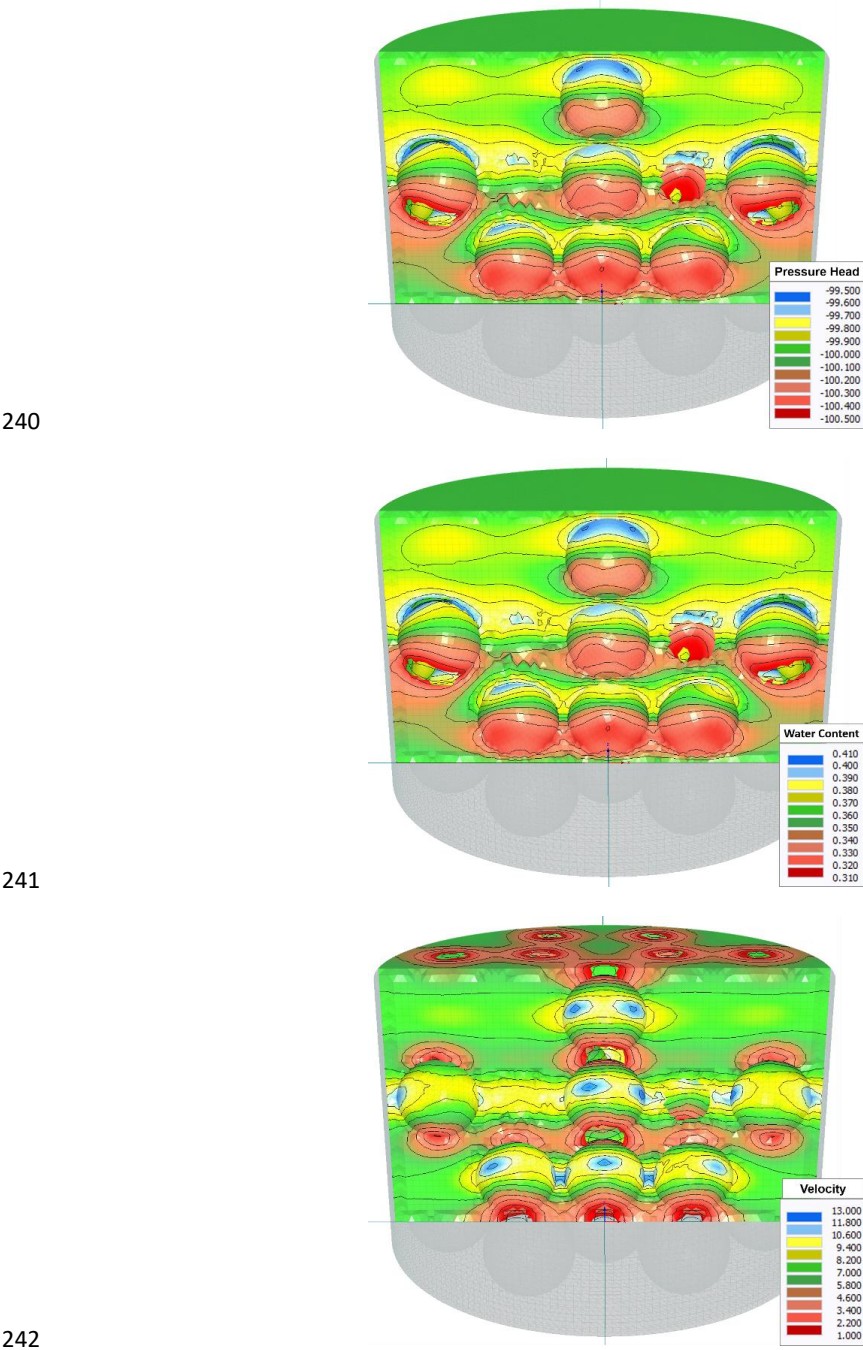




*Figure 2: Visualization of the pressure head (cm), water content (cm³ cm⁻³), and velocity (cm d⁻¹) in a 2D profile in the center of*
*the soil column during the forward simulation of MSUG experiment.*



### 3.2. Comparison of the relative $K_s$ of the scaling models and the 3D simulations under saturated conditions

The dependency of the relative saturated hydraulic conductivity ($K_r$) on the percentage of rock fragments, calculated
by different scaling models and the obtained values from the first pressure in the MSUG experiment is presented in
Fig. 3. The results of the models are shown up to the $f = 37.3$ %, which was the highest value of $f$ simulated in 3D.
However, some of the evaluated models are theoretically valid for higher or lower values of $f$, e.g. 40 % for the Novák
et al. (2011) model and higher values for the GEM model (Naseri et al., 2020).

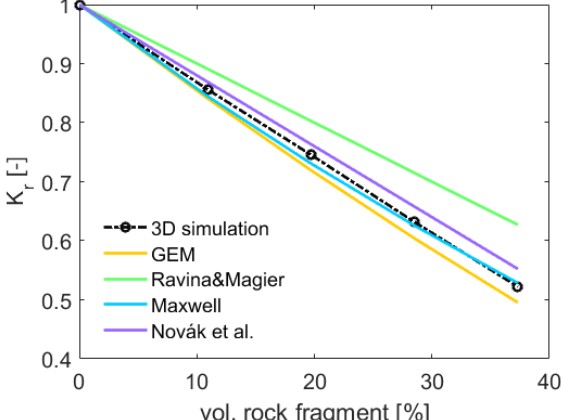


*Figure 3: Comparison of the values of $K_r$ (-) from the MSUG experiment in 3D (circles), and calculated by different scaling models
(solid lines) for volumetric rock fragments up to 37.3 %. The dashed line connects the simulated data points of $K_r$ shown by circles.*

Obviously, the results of our simulations confirm a linear reduction of $K_r$ with increasing volumetric rock fragments
up to $f = 37.3$ % (v/v) in the soil. The numerically obtained values of $K_r$ are shown by circles and connected by the
dashed line in Fig. 3. The dashed line has a slope of -1.29 representing a higher reduction rate of $K_r$ compared to the
scaling of $K_r$ that would be proportional to the volumetric rock fragment content, expressed by Eq. (6) and predicted
by the model of Ravina and Magier (1984) (solid green line). This result supports the fact that even in a stony soil
with spherical impermeable rock fragments, the reduction in the hydraulic conductivity is higher than the reduction
of the average cross-sectional area (which is statistically equivalent to the volumetric fraction of rock fragments).
Hlaváčiková et al. (2016) found an even higher value of -1.45 for spherical rock fragments with a diameter of 10 cm.
The model of Novák et al. (2011) performs better but also leads to a slight under-prediction of the reduction of the





effective saturated conductivity. The performance of this model could be improved by adjusting the parameter $\alpha$ to
match the data of the 3D simulation, but doing this would lead to an unfair comparison with the other models.
The two models predicting a nonlinear relationship between the $K_r$ and $f$, GEM and Maxwell, show similar results at
low contents of rock fragments up to 10 %, with minor differences in outputs of the models. Among all of the evaluated
models of scaling $K_s$, the Maxwell model yields the closest match to the numerically identified values of $K_r$.
We note that these results may differ in natural soils, where an increase of the saturated hydraulic conductivity might
be expected because of macropore flow in lacunar pores at the interface between background soil and rock fragments
(Beckers et al., 2016; Hlaváčiková et al., 2019, Arias et al., 2019). We have not included such a process in our 3D
simulations.
**3.3. Inverse modeling results for effective hydraulic properties**
The observed and fitted time series of the pressure heads at the three representative observation points is shown in
Fig. 4 for the simulated experiments of the four cases with different volumetric rock fragment contents. In each case,
the fitted pressure heads at the three depths of the column (2.5, 5.0 and 7.5 cm) match well with the time series of the
corresponding data from the 3D virtual experiments. Specifically, the match of the pressure heads at 7.5 and 5.0 cm
is excellent, whereas there are slight systematic deviations at the uppermost level at the later stage of the evaporation
experiment.



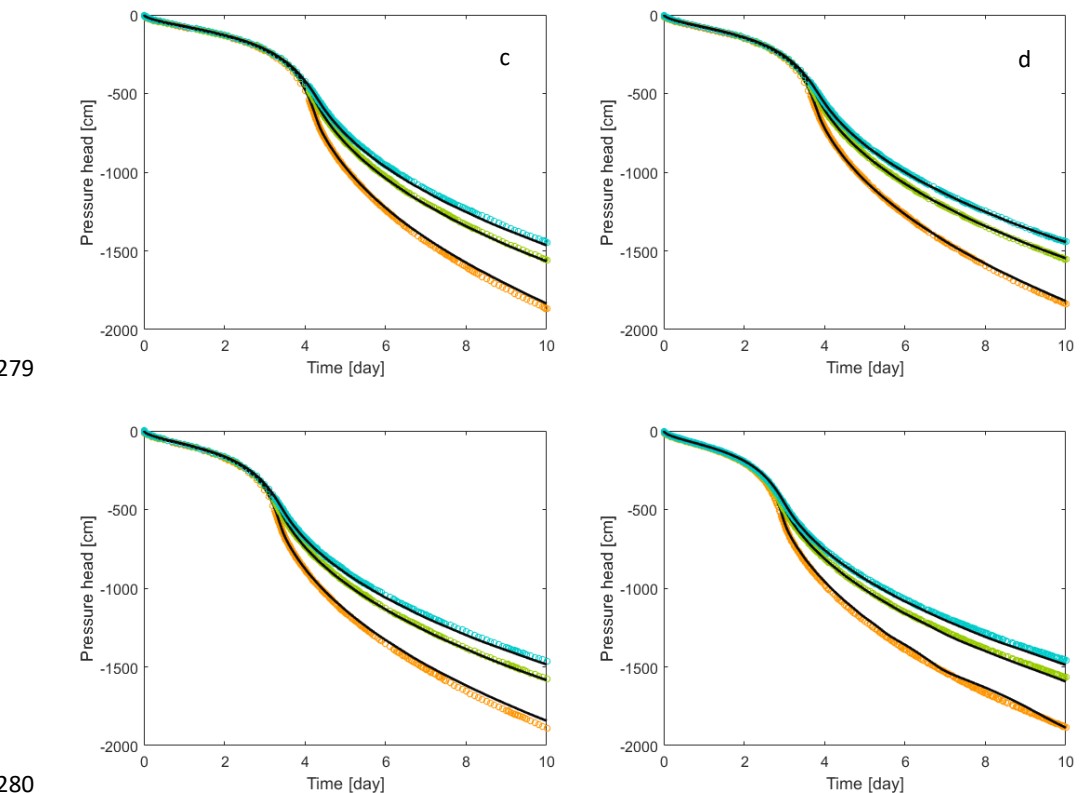



*Figure 4: The time series of the 3D-simulated (circles) and 1D fitted (solid lines) pressure heads at the observation points in three
depths of the stony soil columns with a) 11.0 %, b) 19.8 %, c) 28.5 %, and d) 37.3 % volumetric rock fragment contents (v/v). The
observation depths are indicated by different color codes of orange (upper, 2.5 cm), green (middle, 5.0 cm) and blue (lower, 7.5
cm from top).*


Table 1 shows the values of the root mean square error (RMSE) and mean absolute error (MAE) between the observed
and fitted time series of the pressure heads at three observations points for different contents of rock fragments.
According to the Table 1, the fit is best for the lower rock fragment and in the middle of the column. The highest
deviations occur for the highest rock fragment content but there is no clear trend. Overall, the values of RMSE and
MAE are in an acceptable range regarding the observed values of pressure heads up to -2000 cm. This indicates that
the time series of the pressure heads at multiple depths generated by the 3D simulations of evaporation experiments
can be described successfully by the 1D Richards equation assuming a homogenous system with effective SHP.





*Table 1: The values of RMSE and MAE between the observed and fitted pressure heads, in three observation points for different*
*rock fragment contents.*

| Criteria | Observation point | Volumetric rock fragments (%) | | | |
|---|---|---|---|---|---|
| | | 11.0 | 19.8 | 28.5 | 37.3 |
| RMSE | upper | 12.8 | 6.5 | 12.6 | 18.9 |
| | middle | 4.9 | 2.8 | 5.3 | 10.7 |
| | lower | 9.1 | 3.1 | 8.7 | 13.9 |
| MAE | upper | 10.0 | 4.9 | 9.4 | 14.5 |
| | middle | 4.0 | 2.1 | 4.5 | 8.5 |
| | lower | 7.0 | 2.6 | 6.8 | 11.0 |


The identified SHP are presented in Fig. 5. The solid lines in the Figure show the WRC and HCC of the virtual stony
soils obtained by inverse simulation (except the solid black lines, which are the WRC and HCC of the background
soil), the dashed lines represent the scaled WRC by Eq. (5) and HCC by Eq. (6), and the circles on the HCC plots
represent the discrete data points of hydraulic conductivity obtained by the MSUG. The WRC and HCC are presented
on a $pF$ scale, which is defined as $pF = \log 10 (|h|)$, in which $h$ is the pressure head in cm (Schofield, 1935). The
van-Genuchten model parameters of the background soil and stony soils are shown in Table 2.



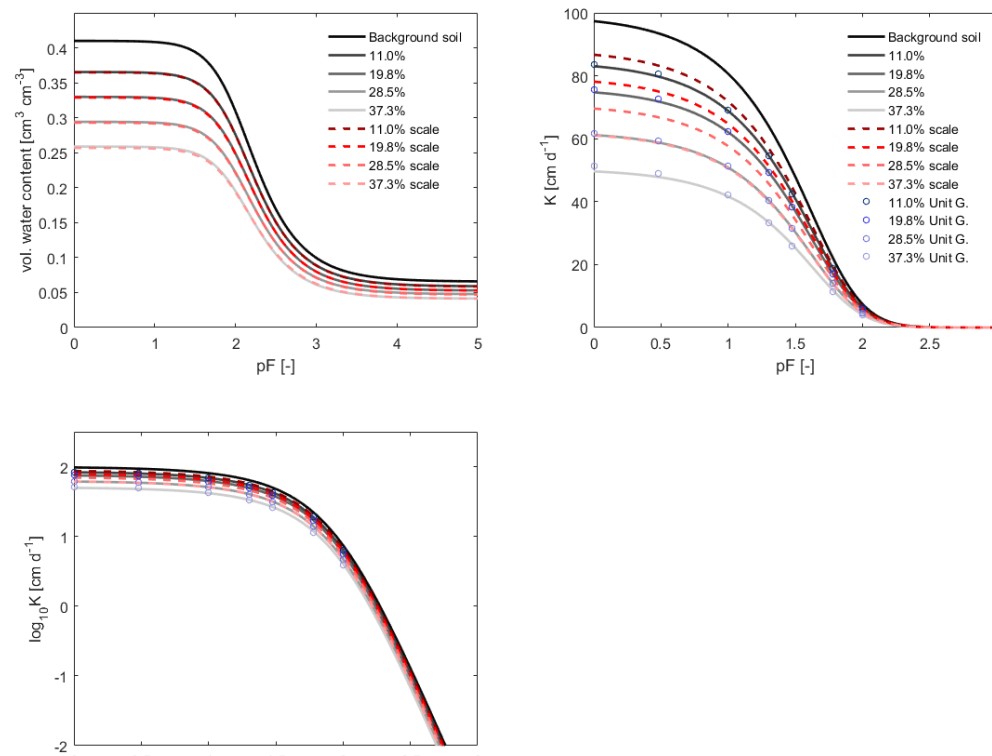



Figure 5: The WRC and HCC of the background soil (solid black line), and identified effective WRC (left) and HCC (right) of the
stony soils (solid gray lines) with different volumetric rock fragments. The HCC are presented also in the logarithmic scale. The
dashed lines show the effective WRC and HCC calculated by the models of Bouwer and Rice (1984) and Ravina and Magier (1984).
The circles on the HCC present the data points of hydraulic conductivity obtained by the MSUG experiment in near-saturated
conditions up to the pF ≈ 2.


Table 2: The van-Genuchten model parameters of the SHP of the background soil and of the inversely determined effective SHP of
the stony soils with different values of $f_c$.

| Parameter | Unit | Volumetric rock fragments (%) | | | | |
|---|---|---|---|---|---|---|
| | | Background soil | 11.0 | 19.8 | 28.5 | 37.3 |
| $\theta_s$ | (cm³ cm⁻³) | 0.410 | 0.365 | 0.330 | 0.294 | 0.259 |
| $\theta_r$ | (cm³ cm⁻³) | 0.065 | 0.059 | 0.053 | 0.048 | 0.041 |
| $\alpha$ | (cm⁻¹) | 0.010 | 0.010 | 0.010 | 0.010 | 0.010 |
| $n$ | (-) | 2.000 | 2.007 | 2.011 | 2.014 | 2.037 |
| $K_s$ | (cm d⁻¹) | 100.0 | 84.7 | 76.2 | 62.3 | 50.4 |
| $\tau$ | (-) | 0.50 | 0.43 | 0.47 | 0.42 | 0.38 |




According to the Fig. 5 and Table 2 the value of the shape parameter $\alpha$ is independent of the rock fragment content
and the change in the value of $n$ is negligible small (but might be systematic). The inversely identified WRC and the
predictions from the Bouwer and Rice (1984) scaling model match almost perfectly for all rock fragment contents. In
agreement with this, there is also an excellent agreement of the values of the saturated ($\theta_s$) and residual water contents
($\theta_r$) (v/v) between scaled and identified WRC. The values of $\theta_s$ and $\theta_r$ in the WRC are directly related to the
volumetric rock fragment contents and the respective values of the background soil and the WRC is scaled by this
factor over the whole range of soil water pressure head.
Similar to the WRC, an increase of the volumetric rock fragments reduces the hydraulic conductivity over the whole
range of pressure head covered by the virtual experiments. However, in contrast to the WRC, the simple scaling model
based on Eq. (6) cannot describe the reduction in HCC. Figure 5 shows that the model of Ravina and Magier (1984,
dashed lines) underestimates the reduction of the effective HCC for all rock contents. The reason might be related to
the local variations of the flow velocity in the soil column. It was shown in Fig. 2 that the variations in the water flow
velocity might be considerable. The nonlinearities in the flow field and changes in the local conductivities, together
with an increased average flow path length, force a stronger overall conductivity reduction. The arrangement of rock
fragments thus might affect the reduction in hydraulic conductivities, leading to different conductivities at the same
volumetric rock fragments (Naseri et al., 2020). The degree depends on how the flow area is altered in the soil column
due to the presence of rock fragments (Fig. 1 and 2). This result is in agreement with Novák et al. (2011) who reported
a higher reduction in conductivity compared to a reduction that is proportional to the rock fragments content.
Furthermore, it may also differ alter depending on the characteristics of rock fragments such as their size, shape and
orientation towards flow (Novák et al., 2011).
We had to include the data points of hydraulic conductivity from the MSUG in the inverse objective function to get a
precise identification of the HCC obtained by inverse modeling near saturation. The information content from the
EVA experiment gives a unique identification only when the flux rate in the system reaches te magnitude of the
unsaturated hydraulic conductivity, which is for many soils around pF 1.5 to to $pF = 2$ (Peters and Durner, 2008).
Although there are some discrepancies visible near saturation for the case with a high value of $f$, the resulting values
of hydraulic conductivity from the MSUG experiment and the inversely identified HCC using the EVA experiment
join well around $pF = 2$ for all of the values of $f$. Therefore, the HCC could be described successfully from the





saturation up to $pF = 3$ using the inverse modeling of the evaporation experiment with added $K$ support points from
the MSUG. The overall results suggest that the effective hydraulic parameters of stony soils could be obtained by the
corresponding real experiments and the result is robust for both, WRC an HCC, even if the uncertainty in the identified
HCC is higher than that of the WRC (Singh et al., 2020; 2021).
**3.4. Evaluation of the Novák, Maxwell and GEM models using the identified HCC**
As stated above, the model of Ravina and Magier (1984) which is a linear scaling approach of the hydraulic
conductivity (Eq. 6) underestimates the reduction of conductivity in the stony soil. We used the identified HCC as a
benchmark to evaluate more advanced models of scaling HCC, namely the Novák, Maxwell and GEM models (Eqs.
7, 8 and 9). Figure. 6 illustrates the calculated HCC of stony soils with different volumetric rock fragments using these
models of scaling HCC and compares them to the identified HCC by the inverse modeling.








*Figure 6: Evaluation of the Novák, Maxwell and GEM models of scaling HCC of stony soils using the identified HCC as a*
*benchmark. The HCC in each case were obtained for different rock fragment contents f=11.0, 19.8, 28.5 and 37.3 % (v/v). The*
*inverse identified curves are shown in solid lines and the model results in dashed lines. The value of $f_c$ in the GEM model was set*
*as 0.982 with the corresponding shape parameter t=1.473 and the parameter $\alpha$ = 1.2 was selected for the Novák model.*





The calculated HCC by the three models are in general in good agreement with the identified HCC in the observed
range of pressure heads. All three models result in a more realistic estimate of the HCC compared to the simple linear
scaling approach. While the model of Novák slightly underestimates the identified HCC for all the four rock fragment
contents, the results are contrary for the GEM model where the reduction in the hydraulic conductivity is
overestimated. The Maxwell model shows the same results as GEM model except for the stony soil with $f =$
37.3 % where it underestimates the HCC.
In order to compare the performance of the three models, the average deviation ($d_{avg}$) between the calculated and
identified HCC (logarithmic scale) was calculated to quantify the error of each model in the pF range 0 to 3 (Table 3).
The signs of numbers in Table 3 represent the tendency of the model in over- or underestimating the identified
hydraulic conductivities. The negative sign means the model underestimates the reduction of hydraulic conductivity.
*Table 3: Performance of the Novák, Maxwell and GEM models quantified by the average deviation of log10 (K) ($d_{avg}$) for*
*different values of rock fragments.*

| model | Volumetric rock fragment (%) | | | |
|---|---|---|---|---|
| | 11.0 | 19.8 | 28.5 | 37.3 |
| Novák | -0.0068 | -0.0028 | -0.0179 | -0.0231 |
| Maxwell | 0.0056 | 0.0162 | 0.0039 | -0.0038 |
| GEM | 0.0077 | 0.0234 | 0.0197 | 0.0248 |


Table 3 confirms the qualitative tendency of underestimation of the conductivity reduction by the Novák model and
the overestimation by the GEM and Maxwell models, but also shows that the difference between the three models is
not large and probably not of relevance in practice (the GEM model at high stone content, which has the highest
deviation, corresponds to a relative mismatch of $K$ of 6 %). However, despite the potential of the three models in
predicting the HCC of stony soils, we think they require further evaluations using field measured data of hydraulic
conductivity in different experimental conditions.
**4. Conclusions**
The objective of our study was to identify the effective SHP of stony soils by inverse modeling of flow experiments
in 3D virtual soils as an alternative tool to laboratory measurements. We were interested to learn if the observed
synthetic data from the forward virtual experiments in such systems with internal structural heterogeneity could be
described by inverse modeling, and if so, whether unique SHP could be identified. We addressed the problem through



detailed 3D numerical simulations of evaporation and multistep unit gradient experiments on 3D stony soils with
different amounts of impermeable spherical rock fragments. The evaporation experiments yielded synthetic data in
the moisture range from saturation to pF 3, which is close to the lower limit of tensiometer measurements in real
experiments. A specific focus of our analysis was on saturated/unsaturated hydraulic conductivity. We identified
effective SHP by inverse simulations with the 1D Richards equation and used them to validate models of predicting
HCC of stony soils. The saturated hydraulic conductivities obtained from the 3D simulations were used as a
benchmark to evaluate and compare the common scaling approaches of the saturated conductivity, $K_s$. Secondly, the
applicability of the scaling models for unsaturated conditions were investigated, namely the Bouwer and Rice (1984)
model of scaling WRC, and Ravina and Magier (1984), Maxwell (Peck and Watson,1979), Novák (Novák et al.,
2011), and GEM (Naseri et al., 2020) models for scaling HCC of stony soils.
The boundary fluxes and the internal system states in the 3D evaporation experiments, represented by the observed
time series of pressure heads at multiple depths, could be matched well by 1D simulations, and the effective WRC
and HCC of the considered stony soils were determined precisely. Comparison with the scaling models showed that
by assuming a homogeneous background soil and impermeable rock fragments, the effective WRC can be calculated
from the WRC of fine soil using a simple correction factor equal to the volume fraction of fine soil, $(1 - f)$. That is
a result with practical implications in obtaining WRC of stony soils. Also, the scaling results for HCC were promising.
Our results confirmed that the reduction in $K_r$ was stronger than calculated by a simple proportionality to $(1 - f)$.
The three models of Novák, Maxwell and GEM consider this and performed adequately in calculating the HCC of the
stony soils. The Maxwell model matched the numerical results best.
Care must be taken before generalizing these results to arbitrary conditions, e.g., highly dynamic boundary conditions
with sequences of precipitation and higher and lower evaporation rates, which might result in different results due to
the occurrence of non-equilibrium water dynamics and hysteresis. For real stony soils, changes in the pore size
distribution of the background soil may result from the presence of rock fragments (Sekucia et al., 2020) with
corresponding consequences for effective SHP. This influence is reported to be more common in compactable soils
with a shrinkage-swelling potential (Fiès et al., 2002). In highly stony soils, where rock fragments are not embedded
completely in the background soil, the existence of effective SHP is still an open question. Finally, the impact of



arrangement and size of rock fragments on evaporation dynamics and effective SHP needs to be understood. Tackling
these problems requires a combination of experimental and modelling approaches.

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
