# Peer review of "Effective hydraulic properties of 3D virtual stony soils identified"

_SOIL, 2021_

## Author Response (AR1)

**Reply to anonymous referee 1**

Understanding the hydraulic properties of stony soils is a valuable research topic. I recommend authors add some laboratory or field-based experimental data to verify the modeling results and strengthen the paper.

Dear referee 1,

We thank you for the time to review the MS and for your recommendation.

Both, investigating synthetic data obtained from numerical process modelling and investigating real measurements is required to enhance our understanding of the water dynamics in soils containing large amounts of rock fragments. The two approaches are complementary and are both necessary to improve our overall understanding of water flow in such systems and of the existence and nature of effective hydraulic properties. Investigating 3D numerical models has the big advantage that we know the underlying structure exactly, that we can vary parameters and properties, and that we can realize a multitude of different variants with limited effort, as e.g. shown by Schlüter et al., 2012 [Schlüter, S., H.-J. Vogel, O. Ippisch, P. Bastian, K. Roth, H. Schelle, W. Durner, R. Kasteel, and J. Vanderborght (2012): Virtual soils: Assessment of the effects of soil structure on the hydraulic behavior of cultivated soils, Vadose Zone Journal 11(3), doi:10.2136/vzj2011.0174.]. This is the approach that we followed in this study.

Real measurements on the other hand, as performed in previous studies of our group in the lab, are laborious and still of limited significance with respect to process understanding. Producing highly reliable data from such systems is a big challenge as e.g. shown by Naseri et al., 2019 [Naseri, M., Iden, S.C., Richter, N. und Durner, W. (2019): Influence of stone content on soil hydraulic properties: experimental investigation and test of existing model concepts, Vadose Zone J. 18:180163. doi:10.2136/vzj2018.08.0163]. Obtaining meaningful measurements from field-based experimental data of soils with high rock fragment contents is even more difficult, and involves a huge effort that requires time, resources, and innovative methods to place sensors in the soil, as illustrated e.g. by Stevenson et al., 2021 [Stevenson, M., M. Kumpan. F. Feichtinger, A. Scheidl, A. Eder, W. Durner P. Blaschke, and P. Strauss (2021): Innovative method for installing soil moisture probes in a large-scale undisturbed gravel lysimeter, Vadose Zone Journal 2021; 1–7. DOI: 10.1002/vzj2.20106].

So, in our opinion both, investigating synthetic data and investigating real measurements, is required to enhance our understanding of the soil water dynamics in soils with appreciable content. However, it cannot be treated all in one single publication.

**Reply to anonymous referee 2**

Dear referee 2,

We would like to thank you very much for reviewing the manuscript and for your constructive feedback. Your comments helped us greatly to improve the article. Below we list our responses and reactions after each paragraph.

The manuscript describes the results of a set of virtual experiments involving the simulation of water flows through a "soil" with different amounts of spherical impermeable fragments, aiming to correspond to coarse fragments, in order to get data that are then used to assess the characteristics of unsaturated water flow and hydraulic conductivity of a "stony" soil compared to what is defined as the "background soil" (the same soil without coarse fragments).

This approach is smart and makes it possible, at the same time, to test the different models proposed to relate these parameters in stony soils vs. their corresponding background soils, and to confirm the already observed fact in other previous studies, that the simple correction suing the proportion of stones is not valid in for hydraulic properties. In this sense, this work adds and supports previous research, and provides solid evidence of the relevance of rock fragments and stones in soil functioning.

In general, my impression is that the manuscript is very well written (both in terms of organization and clarity of the speech), goes to the point without leaving important details

aside, and provides a clear description of the process.

Thank you for sharing your feedback. We appreciate that.

My only concern when reading it was related to the possible limitations of the transposition of the results of this work to real soil and field conditions. Authors have considered this topic at many points in their discussion and description of the approach. In particular, two points seem especially relevant to me, from my experience, deriving from the same fact: that the soil matrix in many cases is not "solid" and can expand and contract when being moistened and dried (and therefore, its total and relative porosity can change with drying), and that the contact between the matrix and soil fragments is not generally at 100% of the fragment surface, with lacunar pores usually existing in the contact. This has consequences in water flows. Authors acknowledge both facts in l.405-407 and 268-271, respectively.

We agree that deviations from our idealized numerical setup will occur in reality and we are aware that natural stony soils will be more complex than assumed. The main factors have been mentioned in your comment. For instance, the freezing/thawing and wetting/drying cycles are the main causes of macropores/cracks and expansive clayey soils are more prone to develop such structural features. The loosening of the contact between fine soil matrix and rock fragments may cause a specific pore space, which we cannot consider in the spatial resolution of our numerical simulation, and which may be furthermore moisture-dependent and variable with time. Further studies are necessary to investigate how these effects propagate into the overall hydraulic behavior. Summing up, you have pointed out some important aspects, which we already referred to in the manuscript.

In this sense, some minor comments that can be done on this ms are

- In some cases, the term "experiment" is used without the qualifier "virtual" (ex. line278). This can be misleading, as in reality no true "experiment" was conducted in this study. This "virtuality" should be clear all along the text, as the conclusions derived from this study are in reality derived from simulations, not actual data.

Thanks for this hint. We agree, and added the phrase throughout the MS to indicate clearly that we investigate synthetic data generated by simulating a virtual soil system. See also Schlüter et al. (2012) on the merits of this approach.

- The discussion on the comparison of the results of the inverse modeling with other models, and in general, with previous studies, has to keep in mind that the data used are based on a set of virtual simulations, whereas it is possible that some of the models were developed based on empirical observations.

Thank you. We will explain and emphasize that in the revised MS. Note that while some of the models are indeed empirical, some have a deeper theoretical foundation.

Other minor comments in relation to the organization of the ms are:

- The Results & Discussion section is usually easier to follow when split into Results and Discussion sections, but in this case I think it is still clear and easy to follow as it is.

Thank you, we agree.

- The conclusions seem more a summary or an abstract in their first half than actual conclusions or take-home ideas of this work. I'd suggest to shorten up this section a bit (or to move the text to the results or abstract section).

That is right. We are going to edit and shorten that section accordingly.